# Effects of Participating in Martial Arts in Children: A Systematic Review

**DOI:** 10.3390/children9081203

**Published:** 2022-08-11

**Authors:** Aleksandar Stamenković, Mila Manić, Roberto Roklicer, Tatjana Trivić, Pavle Malović, Patrik Drid

**Affiliations:** 1Faculty of Sport and Physical Education, University of Niš, 18000 Niš, Serbia; 2Faculty of Sport and Physical Education, University of Novi Sad, 21000 Novi Sad, Serbia; 3Faculty for Sport and Physical Education, University of Montenegro, 81400 Nikšić, Montenegro

**Keywords:** combat sports, physical fitness, motor skills, cardiorespiratory fitness

## Abstract

Background: The application of various martial arts programs can greatly contribute to improving the of physical fitness of preschool and school children. The purpose of this review paper was to determine the effects and influence that martial arts program intervention has on children’s physical fitness, which includes motor skills and the aerobic and anaerobic abilities of children. Method: We searched the following electronic scientific databases for articles published in English from January 2006 to April 2021 to gather data for this review paper: Google Scholar, Pub Med, and Web of Science. Results: After the search was completed, 162 studies were identified, of which 16 studies were selected and were systematically reviewed and analyzed. Eight studies included karate programs, four studies included judo programs, two studies contained aikido programs, and two studies contained taekwondo programs. The total number of participants was 1615 (experimental group = 914, control group = 701). Based on the main findings, karate, judo, taekwondo, and aikido programs showed positive effects on the physical fitness of the experimental group of children. According to the results, the effects of these programs showed statistically significant improvements between the initial and final measurements of most of the examined experimental groups. Conclusion: We concluded that martial arts programs were helpful for improving the physical fitness of preschool and school children, especially for parameters such as cardiorespiratory fitness, speed, agility, strength, flexibility, coordination, and balance.

## 1. Introduction

Martial arts have been practiced for hundreds of years, and today, with modification, they are often used in the form of sports, self-defense, and recreation [1]. Applying well-organized martial arts programs to children can lead to an increase in physical fitness, although the test results are not unequivocal [2]. Many studies have explored the application of various martial arts programs in children in order to improve their motor skills and physical fitness [1,2,3,4,5]. Positive effects could sometimes be visible, especially in children who have been practicing martial arts since childhood. Regarding motor abilities, the development of explosive power, speed of movement, agility, strength, balance, and precision could be noticed [6,7], while on the other hand, in addition to specific physical fitness, aerobic and anaerobic endurance were developed [8,9,10]. In addition, the link between positive socio-psychological responses and involvement in martial arts has also been reported in children. Specifically, increased social skills and self-confidence, along with less aggressiveness, were evident among young martial artists [11]. Furthermore, martial arts can have a significant influence on mental health and on the formation of personal character in young people. Moreover, practicing martial arts helps young people to learn the elements of self-defense and improve their fitness level [12].

The progress of science and technology has led to a decrease in motor activity in favor of a sedentary lifestyle, which has negatively affected the physical development of children [7]. The percentage of children who meet the World Health Organization (WHO) criteria for the recommended level of physical activity [13] is very small and ranges from 2% to 14% and from 9% to 34% in European girls and boys, respectively [5]. Practicing martial arts greatly affects the improvement of the conative characteristics of a person, such as determination, consistency, and motivation [14], as a result of well-organized training, which can directly affect the improvement of physical fitness [15]. This correlates with the results of a study performed by Pinto-Escalona et al. [5]. Pinto-Escalona et al. conducted a one-year investigation in which the experimental group underwent karate intervention at school that led to improved cardiorespiratory fitness and balance in the children. This study was in line with the findings of Pavlova et al. [16] who conducted research with children aged between 5 and 6 years of age. The experimental group (EG) of children underwent a one-year karate program, compared to the classic program of preschool education that the control group (CG) underwent. The results showed that the EG had twice the overall growth rate of physical fitness compared to the CG, with the highest growth rate observed in children in the EG in the “Flamingo” pose (balance). In a study by Boguszewski and Socha [7], which included three groups of children from preschools (children that practiced karate, gymnasts, and inactive children), the best results in the upper extremity strength test were achieved by girls that practiced karate. In addition, based on the obtained results, Boguszewski et al. [17] reported significant improvements in deep squats, shoulder mobility, and torso stability (push-ups) in the experimental karate group. These two karate studies were correlated with the research of Kirpenko et al. [18], which showed that there was a significant improvement in the maximum aerobic capacity, strength, speed–power quality, endurance, and flexibility in the Kyokushin karate experimental group in which children between 10 and 12 years of age were examined. The existing literature reveals the positive impact of various martial arts and combat sports on children’s well-being. The available evidence demonstrates a positive impact on the physiological and physical development of children who practice judo and meet the recommended standards of health-related physical activity [19]. Moreover, Lee and Kim [20] used 16-week taekwondo training in their work and found that children in the the experimental group showed increases in maximum strength and balance. Wasik et al. [21] reported positive effects on body posture; the number of occurrences of body asymmetry was reduced in children and adolescents following a taekwondo training program. In research by Falk and Mor [6], the experimental group of children, which contained children between 6 and 7 years of age, that underwent the 12-week resistance training and martial arts program, showed significant (*p* < 0.05) improvements compared with the control group in the sit-up and long jump tests.

Based on the authors’ findings, there are a small number of review studies that have investigated the effects and impacts of martial arts programs on children’s physical fitness [22]. Considering that the modern age has led to obesity and the deterioration of motor skills in children, this research could contribute to solving this global problem through the organization of martial arts programs, which could be performed as additional activities alongside regular physical education classes. Hence, this review aimed to determine the effects and influence that martial arts program intervention has on children’s physical fitness, which includes motor skills and the aerobic and anaerobic abilities of children.

## 2. Materials and Methods

### 2.1. Inclusion Criteria

The following criteria were used to select the articles to be included in the review: (1) original scientific articles; (2) all studies that dealt with martial arts; (3) articles written in English; (4) articles where the participants in the sample were children of preschool and school age (between 4 and 18 years of age).

### 2.2. Exclusion Criteria

The following criteria were used to select the articles to be excluded from the research: (1) review studies; (2) studies in which the subjects were children younger than 4 years of age and older than 18 years of age; (3) articles with research in which the experimental groups of children were not subjected to martial arts training programs; (4) articles in which the results or investigated parameters were not adequately presented for further analysis.

### 2.3. Search Strategy and Study Selection

Electronic database searches were performed using Google Scholar, Pub Med, and Web of Science. The search terms covered the areas of the influence of martial arts training on physical fitness and martial arts programs for preschool and school children using a combination of the following keywords: “martial arts”, “physical fitness”, “children”, “motor skills”, “cardiorespiratory fitness”. The results of a search of articles written in English and published between January 2006 and April 2021 were analyzed. Articles from the database list that were clearly not relevant were removed before assessing all other titles and abstracts using our predetermined inclusion and exclusion criteria. Inter-reviewer disagreements were resolved by consensus opinion or arbitration by a third reviewer. Reference lists of the selected manuscripts were also examined for any other potentially eligible articles.

### 2.4. Assessment of Bias

Two independent reviewers assessed the risk of bias. The agreement between the two reviewers was evaluated using k full-text screening statistics and an assessment of relevance and the risk of bias. When there was disagreement over the risk of bias, the third reviewer checked the data and formed the final decision. The agreement rate for k among reviewers was k = 0.95. Inter-observational reliability and agreement were calculated using the interclass correlation coefficient and Cohen’s kappa test with values interpreted as follows: 0—no agreement; 0.01–0.20—slight agreement; 0.21–0.40—good agreement; 0.41–0.60—moderate agreement; 0.61–0.80—significant agreement; and 0.81–1.0—almost perfect agreement.

## 3. Results

### Selection and Inclusion of Studies

We collected 159 studies through a database search in Google Scholar, PubMed, and Web of Science. Three more studies were found from other sources; thus, the total number of collected studies was 162. Next, we excluded 63 duplicate studies (duplicate search or publication of the thesis in a journal). After reviewing the research titles and abstracts of the 99 remaining selected studies, an additional 26 studies were excluded that included children with diseases and children older than 18 years of age. Another 10 studies were excluded because they included martial arts that were not classified as budo martial arts. In addition, seven other studies were excluded because full texts were not available. Therefore, we selected 67 studies after excluding 32 studies. We reviewed the full text of the 67 selected studies and classified them according to the PICOS criteria. After that, 24 studies with no control groups, seven studies with unclear statistics, and 18 studies with uncertain training durations were excluded. Therefore, we finally included 16 studies (8 studies included only male participants, 1 study included only female participants, and 7 studies included both sexes) (Table 1). The data selection method is presented within the following flow diagram (Figure 1).

The study included 16 original research articles that tested the effects of martial arts programs in children aged between 4 and 13 years of age. The articles are presented in tabular form by year of publication from 2006 to 2021, respectively. There were no control groups in four articles [2,23,30,32]. The youngest participant was 4.5 years old [7], and the oldest participant was 13 years [33]. The total number of participants was 1615 (EG = 914, CG = 701). The sample sizes of the studies were analyzed and ranged from 10 [25] to 721 children [5]. In seven studies, both genders were included [2,4,5,7,25,30,31]. In eight studies, the participants were only male children [16,18,23,26,27,28,29,32], while one study included only girls [24]. The longest experimental treatment lasted two years [18] and the shortest only lasted a week [26]. Frequently, the duration of the sessions in all martial arts programs was 60 min [2,4,5,7,16,26,27,28,32]. Eight studies included experimental treatments of karate [2,4,5,7,16,18,26,29], four studies included judo treatments [23,24,25,32], in two articles taekwondo treatments were employed [26,27], and two studies included experimental aikido treatments [27,28].

## 4. Discussion

This study examined the effects of various martial arts programs on children’s physical fitness. Based on the main findings, karate, judo, taekwondo, and aikido programs showed positive effects on physical fitness components. According to the results, the effects of these programs showed significant differences between the initial and final measurements of most of the examined experimental programs, but also when compared to the control groups. Cardiorespiratory fitness, speed, agility, strength, flexibility, coordination, and balance were used to assess physical fitness, while other parameters, such as body composition, mental conative, and cognitive capacities, were excluded.

### 4.1. Cardiorespiratory Fitness

Cardiorespiratory fitness as a parameter of physical fitness was represented in only three studies [18,28,32]. After a 24-month karate program, Kyrpenko et al. [18] found a large statistically significant improvement (*p* < 0.01) in the cardiorespiratory endurance parameter in the EG of karatekas, while in inactive children in the CG, this was not the case. The 1000 m test is a middle-distance running test that gives participants the opportunity to increase their cardiorespiratory fitness at the expense of muscle strength and to improve their running technique, unlike running for 20 m, 30 m, and 100 m, where the type of muscle fibers, despite the training, plays a major role. Since the treatment in the EG lasted 24 months, it was no wonder that there was a significant improvement in the final measurement, given that the subjects had enough time to increase their strength and improve their running technique. In addition, Brasil et al. [32] assessed the cardiorespiratory ability of obese and non-obese children after participating in a 12-week judo program. The authors concluded that there was a decrease in the VO2peak parameter in the obese children and statistically significant differences in HR at the VO2peak between non-obese and obese children. Given that obese children have an excess of inefficient adipose tissue that leads to accelerated fatigue, it was no wonder that HRs at the VO2peaks were higher in the obese children. Pop et al. [28] examined cardiorespiratory endurance, where the EG, in addition to regular classes of physical education, was subjected to an eight-month aikido program, while the control group attended only physical education classes. Testing was performed using the 20 m beep test, and, on that occasion, it was concluded that there were no statistically significant differences between the EG and the CG in the final measurements.

### 4.2. Speed and Agility

The shuttle run agility test has been the main tool to test this motor ability in many studies in which agility as a parameter of physical fitness was presented [7,18,23,24,29,31]. Specifically, using the nine-month judo program, Sekulic et al. [23] found statistically significant differences (*p* < 0.05) in the final measurements of the 4 × 1.98 m shuttle run agility test in an experimental group of subjects composed of boys. This was in correlation with the findings of Krstulović et al. [24], who also found statistically significant differences using the same test as the judo program of the same duration in female participants. In addition, using a six-month karate program, Boguszevski and Socha [7] used the 4 × 5 shuttle run test to assess agility, concluding that progress had been made in the final measurements, but they remained statistically insignificant. This was not correlated with Ma and Qu [29], who used the same 4 × 5 shuttle run test to identify the effects of a two-month karate program. They found a statistically significant difference (*p* < 0.01) in the final measurement of the EG that correlated with the results obtained by Top et al. [31] and Kyrpenko et al. [18], who tested the agility of the 4 × 9 shuttle run test using a two-year karate program and found a statistically significant improvement (*p* < 0.01) in the EG compared to the CG.

Padulo et al. [26] used frontal/side jumps in a quadratic agility assessment test to compare the effects of a one-week high- and low-intensity karate program. They found a statistically significant improvement in the EG. They also found a statistically significant difference between the EG and the CG in favor of the EG on the final measurement. Speed was one of the monitored parameters of physical fitness in five articles [4,23,24,25,28]. Various running speed tests were used to measure speed as a parameter of physical fitness. Sekulic et al. [23] concluded that the change in speed in boys did not occur in the initial and final EG measurements. This correlated with the findings of Krstulović et al. [24], who applied almost the same program for 9 months in female participants; however, this was not correlated with the findings of Demiral [25], whose research showed statistically significant differences between the EG and the CG in the 20 m sprint test. Perhaps the reason was that Demiral’s study lasted 3 months longer than the previously mentioned studies [23,24], where the participants were subjected to experimental treatment for a longer period of time. A study conducted by Pop et al. [28] used 20 m sprint tests, while Alesi et al. [4] applied a transfer-running test to check the speed skill of participants and also found statistically significant differences between the EG and the CG in favor of the EG.

### 4.3. Strength

Strength, as a parameter of physical fitness, was assessed in 12 research articles [2,4,7,16,18,23,24,25,26,29,30,31]. In most of the studies, the explosive strength of the legs and arms was measured using standing long jumps and medical ball throws tests. Specifically, in all studies in which explosive power was tested by these test protocols, statistically significant improvements in the EGs were present [2,4,7,16,18,22,23,24,25,26,29,31] and also in some CGs [16,22,24]. Since the ULES parameter (upper limb explosive strength) was presented in five articles [7,16,25,26,29], and the fact that in each of these articles medical ball throws were used, it was the main test for estimating this parameter. In four of the five studies, the karate program was applied, while the judo program was implemented in only one study [25]. Considering that karate is a striking martial art and sport, which requires highly developed explosiveness and quickness that is constantly emphasized during the training, it is absolutely justified and logical that this parameter improved. Repetitive strength was presented in three studies [7,16,29]. Based on the results of their research, Boguszevski and Socha [7] found no statistically significant differences between the EG and the CG on the initial and final measurements of the sit-up test in their study in which a six-month karate program was applied. This correlated with the results obtained by Ma and Qu [30], who tested the repetitive strength of karatekas conducting the same test and concluded that there were no statistically significant improvements in their EG at the final measurements. Interestingly, the authors of both articles came to the same conclusions, even though the karate program proposed by Boguszevski and Socha [7] lasted 24 weeks and the program of Ma and Qu [29] lasted only 8 weeks. It is probable that the main reason for this is the fact that there were no exercises in the experimental karate programs that affected the repetitive strength of the abdominal muscles. Pavlova et al. [16] found that after a 12-month karate program, there was a statistically significant improvement (*p* < 0.01) in the EG in repetitive leg strength. A significant improvement (*p* < 0.01) was also registered in the CG, where the participants attended only physical education classes. Since the children were aged between 5 and 6 years old at the beginning of the treatment, it is likely that one of the reasons for the great improvement was maturation.

### 4.4. Flexibility

Flexibility was one of the parameters for assessing physical fitness in 10 articles [2,5,7,18,23,24,26,27,29,30]. The most common flexibility assessment test used in as many as five articles was the “sit and reach” test [2,23,24,29,30]. The results of this test showed statistically significant differences in flexibility between the EG and the CG in the final measurements in two articles by Krstulović et al. [24] and Sekulic et al. [23], in which 9-month judo programs were conducted. This was partially in line with the findings of Rutkowski et al. [2], who, by applying a 10-week karate program, found a statistically significant decrease in flexibility in normal-weight boys, while in normal-weight girls, there was a statistically significant difference. In addition, this did not correlate with the results of Pathare et al. [30] and Ma and Qu [29]; in these studies no statistically significant improvement in flexibility was observed. In addition, the author Mrockowski [27] used a nine-month aikido program, where the hip flexibility test “samurai walk” was used to examine flexibility, and the presence of statistically significant differences between the EG and the CG were evident. Boguszevski and Socha [7] used the “finger floor” test to assess flexibility and found that the participants in the EG group (girls) significantly improved their performances compared to the control group. This correlated with Padulo et al. [26], who obtained similar results, checking the flexibility of the joints after a one-week program of high-intensity karate (EG) and low-intensity karate (CG); however, this did not correlate with the findings of Kirpenko et al. [18] who concluded that 12 months of karate did not cause an improvement in flexibility.

### 4.5. Coordination

Coordination as a parameter of physical fitness was represented in four articles [23,24,25,31]. The results of the 10 m polygon test conducted by Krstulović et al. [24] and Sekulić et al. [23] did not show statistically significant differences in coordination between the EG and the CG in the final measurements of both studies. In his research, Demiral [25] applied the coordination test (balance skill) and rapidity test in the EG and the CG of judokas and found statistically significant improvements (*p* < 0.01) in both sexes of the EG. In addition, statistically significant differences (*p* < 0.01) in coordination were found in the EG of girls compared to the CG. However, these results were partially in line with the results of Top et al. [31], which indicated a statistically significant improvement (*p* < 0.05) in the EG, but no statistically significant differences between the EG and the CG in the final measurements were observed.

### 4.6. Balance

Balance was tested in the majority of studies using the “flamingo” balance test [16,25,28]. In addition, other articles used the following tests: the Y balance test [5], the single-leg balance test with closed and open eyes [2,31], and a test that involved balancing on a force plate device [30]. Demiral [25] found a statistically significant improvement in balance in the EG of boys and girls following a 12-month judo program. These findings were not correlated with the results of Pop et al. [28], who found no significant differences between the EG and the CG after an eight-month aikido program in addition to regular physical education classes. In addition, according to Demiral’s findings, a partial correlation was found with the study of Pavlova et al. [16]; the EG achieved a statistically significant improvement in balance, which was the same as the CG, whose participants only attended regular physical education classes. Speaking of the Y balance test, Pinto-Escalona et al. [5] found a slight improvement in the EG group that was not statistically significant. In the research performed by Rutkowski et al. [2], in which all the subjects underwent experimental treatment, only boys of normal weight achieved a statistically significant improvement in balance. This was not consistent with the findings of Pathare et al. [30], who discovered that overweight children had greater improvements in balance than healthy-weight children.

### 4.7. Future Research

Important considerations were identified from this review to support the development of future research in this research field, such as the following:One needs to be cautious when adopting an intervention program model that has already been performed and has relevant methodological limitations.There is a lack of studies dealing with the impact of martial arts on children’s cardiorespiratory and motor-skill parameters. In future studies, the emphasis should be on non-budo martial arts such as capoeira, muay Thai, Brazilian jiu-jitsu, wrestling, and other combat sports [33].The following studies might have investigate preschool children because, in most countries, children start practicing martial arts from an early age.Various martial arts programs for school and preschool children need to be analyzed and a larger number of physical fitness parameters with an emphasis on body composition, as well as the heterochronism of their development, need to be monitored.

### 4.8. Strengths and Limitations

This is one of the first review papers that combined several different martial arts and analyzed their effect on children, including a comprehensive search of studies and the assessment of their methodological qualities. The main limitation of this study is the fact that, in addition to judo, taekwondo, and aikido, the majority of the research papers were based on karate. Therefore, the karate program was the most effective in terms of positive impact on physical fitness. Additionally, a significantly larger number of male children were included in the monitored studies. Given that there are large differences in motor skills and cardiorespiratory fitness between boys and girls, the gender should be defined separately.

## 5. Conclusions

This review confirmed that martial arts programs lead to improved physical fitness in preschool and school children. Based on the results of the analysis, it could be concluded that cardiorespiratory fitness, speed, agility, strength, flexibility, coordination, and balance were the most important parameters of physical fitness, which demonstrated considerable improvement in the final measurements of EG participants. The obtained information could be very useful in promoting and advertising the positive aspects of these budo martial arts, which can directly affect children’s and parents’ choices when it comes to choosing the sport that they will practice.

## Figures and Tables

**Figure 1 children-09-01203-f001:**
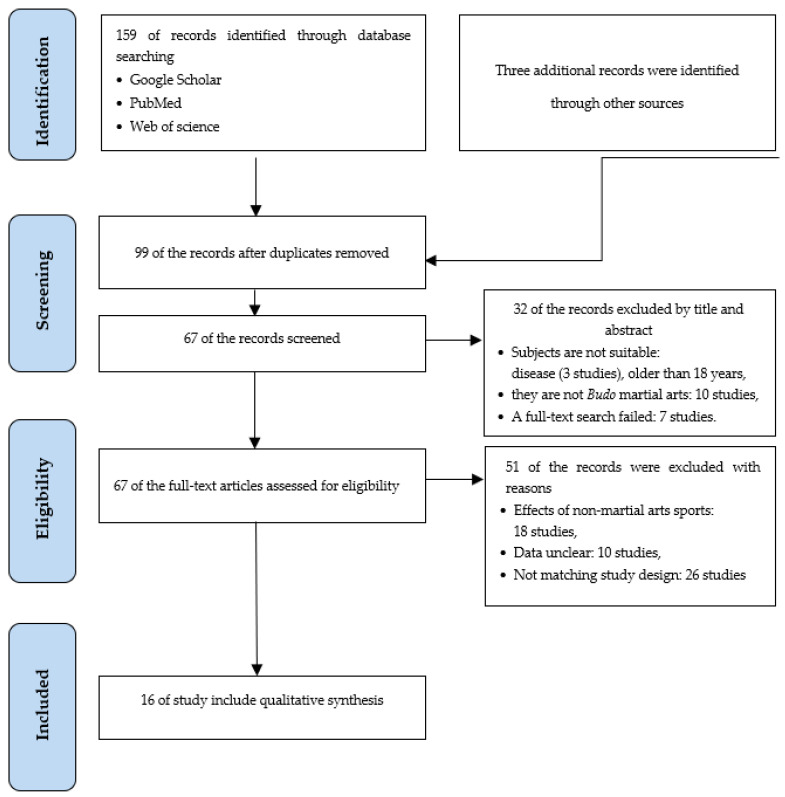
PRISMA flow diagram.

**Table 1 children-09-01203-t001:** Results of the included studies.

Author (Year)		Subjects Program					Intervention		
	Age and Gender (y)	E (*n*)	C (*n*)	Duration (wk/m)	Frequency (Sessions/ (wk/)	Time (min/ Session)	E Training Content	Testing Content	Results after Intervention
Sekulic et al. (2006) [23]	7, M	Judo (*n* = 41)	Sports games (*n* = 57)	9 m	3	45	Judo training	COR (10 m polygon test); AGL (4 × 1.98 m shuttle run test); FLEX (maximal circumduction, sit and reach); SP (20 m sprint); LLES (standing long jump); CREN (3 min run); MEN (flexed-arm hang, 1 min of sit-ups).	All variables (E, C) ↑*; AGL, E↔*; FLEX, E↔*; MEN, E↔*; COR, (E, C) ↔.
Krstulović et al. (2010) [24]	7, F	Judo (*n* = 30)	Sports games (*n* = 49)	9 m	3	45	Judo training	COR (10 m polygon test); AGL (4 × 1.98 m shuttle run test); FLEX (maximal circumduction, sit and reach); SP (20 m sprint); LLES (standing long jump); CREN (3 min run); MEN (flexed-arm hang, 1 min of sit-ups).	All variables (E, C) ↑*; AGL, E ↔*; FLEX, E ↔*; MEN, E ↔*; MS, E ↔*; COR, (E, C) ↔.
Boguszevski and Socha (2011) [7]	4.5–6.5, M and F	Karate (*n* = 30)	Corrective gymnastics (*n* = 30); inactive (*n* = 28)	6 m	3	60	Karate training	RS (20 s of sit-ups); ULES (throwing 1 kg medicine ball); LLES (long jump from the start line); AGL (4 × 5 m shuttle run); FLEX (test “fingers-floor”).	ULES, E (G) ↑*, E(B) ↑; LLES, E (G) ↔*; RS, ↔; AGL, E (G) ↑; FLEX, E (G) ↔*.
Demiral (2011) [25]	7–12, M and F	Judo, (*n* = 38),	Judo (*n* = 31)	12 m	ND	ND	Judo training	BAL (“Flamingo” balance test); COR (coordination test, rapidity test); LLES (standing long lump); ULES (medical ball throws); SP (running speed test); S (grip and back power test).	LLES, E (B, G) ↑*; (B, G) ↔*; ULES, E (B) ↑*; BAL, E (B, G) ↑*; COR, E (B, G) ↑*, G ↔*; SP, E (B, G) ↑*, B ↔*; S, E (B, G) ↑*.
Alesi et al. (2014) [4]	9–10, M and F	Karate (*n* = 19)	Inactive (*n* = 20)	8 wk	3	60	Karate training	AGL (coordination skills with hurdles); LLES (standing broad jump); SP (20 m sprint).	AGL, (E, C) ↔*; LLES, (E, C) ↔*; SP, (E, C) ↔*.
Padulo et al. (2014) [26]	8–12, M	High-intensity karate (*n* = 53)	Low-intensity karate (*n* = 20)	1 wk	14	60	Karate training	ULES (sitting medicine ball throw); LLES (standing long jump); AGL (frontal/lateral jumps); FLEX (“active” joint flexibility).	ULES, (E, C) ↔*, C ↑; LLES, (E, C) ↔*, E ↑*, C ↑; AGL, (E, C) ↔*, E ↑*, C ↑; FLEX, (E, C) ↔*, E ↑*, C ↑;
Mroczkowski (2015) [27]	7–10, M	Aikido and PE classes (*n* = 66)	PE classes (*n* = 41)	9 m	3	60	Aikido training	FLEX of hip *(*“samurai walk”).	FLEX c1-TAOR (Lh), (E, C) ↔*, E ↑*; c2- AOER (Lh), E ↑; c3- AOIR (Lh), (E, C) ↔*, (E, C) ↑*; c4- TAOR (Rh), (E, C) ↔ *, E ↑*; c5- AOER (Rh), E ↑*; c6- AOIR (Rh), (E, C) ↑.
Pop et al. (2017) [28]	9–10, M	Aikido and PE classes (*n* = 5)	PE classes (*n* = 5)	8 m	3	60–90	Aikido training	BAL (“Flamingo” balance test); CREN (beep test, 20 m); SP (TRANSFER running 10 × 5 m at speed).	BAL, (E, C) ↔; CREN, (E, C) ↔; SP, (E, C) ↔*.
Ma and Qu, (2017) [29]	8–12, M	Karate (*n* = 51)	*/*	8 wk	7	5	Karate training	RS (30 s of sit-ups); LLES (standing long jump); ULES (throwing a medicine ball); AGL (4 × 5 m shuttle run); FLEX (sit and reach).	LLES, ↑*; RS, E ↑; AGL, ↑*; FLEX, ↔.
Pavlova et al. (2018) [16]	5–6, M	Karate (*n* = 33)	PE classes (*n* = 38)	12 m	3	45–60	Karate training	LLES (standing long jump); ULES (medical ball throws); EN (jumping with a rope to fatigue); RS (max. number of squats to fatigue); BAL (“flamingo” balance test).	LLES, (E, C) ↑*; ULES, E ↑*; RS, (E, C) ↑*; BAL, (E, C) ↑*; EN, (E, C) ↑*.
Pathare et al. (2018) [30]	6–12, M and F	Taekwondo HW (*n* = 11), OW (*n* = 6)	*/*	10 wk	2	50	Taekwondo training	LLES (just jump system, vertical jump); BAL (force plate (Watertown, Mass.) bipedal/normal stance eyes open and closed, single stance eyes open (30 s)); FLEX (sit and reach test); PH activity (Yamax SW-701 pedometer).	LLES, (HW, OW) ↔*; PH activity, (HW, OW) ↔; LLES, (HW, OW) ↔; BAL, (OW, HW) ↔, OW ↑; FLEX, (HW, OW) ↓*, ↔.
Top et al. (2018) [31]	7–10, M and F	Taekwondo B (*n* = 13), G (*n* = 9)	Inactive B (*n* = 12) G (*n* = 8)	12 wk	3	60	Taekwondo training	S (knee push-ups, sit-ups, wall sit); COR (touching nose with a finger); LLES (standing long jump); BAL (single leg balance test with closed and open eyes); AGL (shuttle run, stepping sideways, one-legged stationary hop, one-legged side hop, and two-legged side hop).	COR, (E, C) ↔, E ↑*; S, LLES, (E, C) ↔*, E ↑*; AGL, (E, C) ↔*.
Rutkowski et al. (2019) [2]	7.6 ± 0.4 M and F	Karate B/N (*n* = 15), G/N (*n* = 15), B/O (*n* = 15), G/O (*n* = 14).	*/*	10 wk	2	60	Karate training	EUROFIT FITNESS TEST: BAL; Arm SP; Abdominal MS; LLES; FLEX (sit and reach test).	BAL, B/N ↑*; Arm SP, B/N ↑*; (G/N, G/O) ↔*, (B/N, B/O) ↔*; (G/O, B/N) ↔*; Torso MS; (G/O, B/O) ↔*, G/O ↓; LLES, G/N ↔*; FLEX, B/N ↓*, G/N ↔*.
Brasil et al. (2020) [32]	8–13, M	Judo (*n* = 35), 20 Ob, 15 NOb.	*/*	12 wk	2	60	Judo training	CPET treadmill: 1. VO2peak; 2. HR at VO2peak.	VO2peak, E (Ob) ↓; HR at VO2peak, E ↔*.
Kyrpenko et al. (2020) [18]	10–12, M	Karate (*n* = 27)	Inactive (*n* = 29)	24 m	ND	ND	Karate training	CREN (1000 m running); Arm S (pull-up on the crossbar); LLES (long jump from place); AGL (4 × 9 m shuttle run); FLEX (body leaning forward); VO2max	CREN, E ↑*; LLES, E ↑*; AGL, (E, C) ↑*; FLEX, ↔; VO2max, E ↑*.
Pinto-Escalona et al. (2021) [5]	7–8, M and F	Karate (*n* = 388)	PE classes (*n* = 333)	12 m	2	60	Karate training	CREF (20 m shuttle run test); FLEX (frontal split test); BAL (Y balance test).	CREF, E; FLEX, E; BAL, E.

Legend: M—male; F—female; E—experimental group; C—control group; wk—week; m—month; ND—not defined; PE—physical education; B—boys; G—girls; B/N—boys of normal weight; G/N—girls of normal weight; B/O—overweight boys; G/O—overweight girls; Ob—obese; NOb—non-obese; PH—physical; HW—healthy weight; OW—overweight; S—strength; MS—muscle strength; ES—explosive strength; ULES—upper limb explosive strength; LLES—lower limb explosive strength; RS—repetitive strength; COR—coordination; AGL—agility; FLEX—flexibility; SP—speed; BAL—balance; EN—endurance; MEN—muscular endurance; CREN—cardiorespiratory endurance; CREF—cardiorespiratory fitness; FMS—functional movement screen; CPET—cardiopulmonary exercise testing; TAOR—total angle of rotation; AOER—angle of external rotation; AOIR—angle of internal rotation; Lh—left hip; Rh—right hip; ↑—increased (↑*– significantly increased); ↓—decreased (↓*– significantly decreased); ↔—no significant difference; ↔*—significant difference; (E, C)—between groups E and C.

## Data Availability

Data are contained within the article.

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
