# Peer review of "Effects of Participating in Martial Arts in Children: A Systematic Review"

_children, 2022, doi:10.3390/children9081203_

Round 1
Author Response
Thank you for your comments and suggestions. All corrections have now been made as noted.
Reviewer 2 Report
I have included my comments and suggestions in the attached file.

Author Response
Thank you for your useful comment. The introduction has now been slightly expanded, and some of the suggested studies have been included (four).
Reviewer 3 Report
Thank you for permitting me to review this manuscript
The aim of the review in the abstract and in the inroduction differs
it is written in the abstract
This review paper aims to analyze the collected research and determine the impact of different martial arts programs on the physical fitness of preschool and school-age children.
while in the introduction it is written
Hence, this review aimed to determine the effects and influence of martial arts program intervention on children's physical fitness, which 80 includes motor skills and aerobic and anaerobic abilities of children.
in the abstract version the reader may think that this review was designed to assess the effect of each martial art in the children while in the introduction bersion (which is appropiate , it is the effect of all martial art ) therefore please rephrase in the abstract
Author Response
Thank you for your comment. The abstract is now rephrased accordingly.
Round 2
Reviewer 2 Report
Yes, the article has been corrected, supplemented and is now worthy of approval for publication.